# High-Frequency and High-Resolution VLBI Observations of GHz Peaked Spectrum Objects

**Xiaopeng Cheng** [1,*] , **Tao An** [2,3,*] , **Ailing Wang** [2,4] and **Sumit Jaiswal** [2]

1    Korea Astronomy and Space Science Institute, 776 Daedeok-daero, Yuseong-gu,
     Daejeon 34055, Republic of Korea
2    Shanghai Astronomical Observatory, Chinese Academy of Sciences, Nandan Road 80,
     Shanghai 200030, China
3    Xinjiang Astronomical Observatory, Chinese Academy of Sciences, Urumqi 830011, China
4    University of Chinese Academy of Sciences, 19A Yuquanlu, Beijing 100049, China
*    Correspondence: xcheng@kasi.re.kr (X.C.); antao@shao.ac.cn (T.A.)

**Abstract:** Observational studies of GHz peaked spectrum (GPS) sources contribute to the understanding of the radiative properties and interstellar environment of host galaxies. We present the results from the multi-frequency high-resolution VLBI observations of a sample of nine GPS sources at 8, 15, and 43 GHz. All sources show core-jet structure. Four sources show relativistic jets with Doppler boosting factors ranging from 2.0 to 5.0 and a jet viewing angle between $10°$ and $30°$. The core brightness temperatures of the other five sources are below the equipartition brightness temperature limit with their jet viewing angles in the range of $13.6°$ to $71.9°$, which are systematically larger than those of relativistic jets in this sample. The sources show diverse variability properties, with variability levels ranging from 0.11 to 0.56. The measured turnover frequency in the radio spectrum ranges from 6.2 and 31.8 GHz (in the source's rest frame). We estimate the equipartition magnetic field strength to be between 9 and 48 mG. These results strongly support the notion that these GPS sources are young radio sources in the very early stage of their evolution.

**Keywords:** galaxies; active; galaxies; evolution; radio continuum; galaxies

## 1. Introduction

Observational study of GHz-Peaked-Spectrum (GPS) sources is crucial for advancing our understanding of the evolution of radio sources [1]. GPS sources are thought to be young radio galaxies that are still in the early stages of their evolution [2,3]. By studying GPS sources at different redshifts, we can trace their evolution over time and gain insights into the processes that drive the development of radio sources.

Observations of radio-loud active galactic nuclei (AGN) in high-redshift space show a high incidence of peaked-spectrum sources (close to 50% in the flux density-limited sample) [4–6], much higher than in a low-redshift radio source sample.

The study of high-redshift GPS sources is important for a number of reasons: (1) understanding the birth and growth of radio galaxies in the early universe. (2) Probing the interstellar medium. GPS sources are often surrounded by dense clouds of gas and dust [7,8], which can obscure the central regions of the source. By studying these clouds, we can learn more about the interstellar medium and the conditions that prevailed in the early universe. (3) Testing models of black hole and galaxy growth. GPS sources are thought to have powerful radio jets that play a key role in shaping the structure of surrounding galaxies and influencing the evolution of their host galaxies [9–11]. We can gain important insights into the physics of AGN and the role of radio jets in the early universe.

The distinguishing observational features of GPS sources are a compact radio structure ($\lesssim$1 kpc) and a convex radio spectrum peaking at ~1 GHz (in practice, the turnover frequency is between 0.5 and 10 GHz) [1,10]. The compact nature of these objects is

thought to be either from frustrated sources where the jet is confined by the surrounding dense medium in their host galaxy environments (e.g., [8,12]), or they are young radio sources (age $\approx 10^2$–$10^5$ yr) which are still growing (e.g., [2,3]). In the second scenario, young sources with sustained long-term nuclear activity may evolve into large-size radio galaxies through a medium symmetric object (or compact steep-spectrum source) stage [3]. However, the growth of the frustrated sources is stagnated by the dense interstellar medium (ISM) and unable to escape [8]. High-quality high-resolution VLBI images can resolve the radio structure of GPS sources and study the radio properties of the individual components (e.g., spectral indices, proper motions, Doppler boosting), providing an opportunity to distinguish between the two scenarios.

Synchrotron spectral aging is another approach for evaluation of the source age based on the determination of the turnover of the synchrotron self-absorption spectrum, which has a spectral break at progressively lower frequencies over time [13–16]. Based on the classical theory of synchrotron radiation, three models have been developed: one by Kardashev [13] and Pacholczyk [15], referred to as the KP model; one by Jaffe & Perola [16], called the JP model; and one by Kardashev [13], known as continuum injection (CI) mode. The KP and JP models assume a single injection of relativistic particles, which need to be distinguished from different regions. The CI mode, on the other hand, assumes a continuous injection of relativistic electrons over the lifetime of the source, in which these regions are not spatially resolved.

Recently, we reported high-resolution images of a sample of 134 compact AGNs observed with the VLBA at 43 GHz, resulting in the highest resolution of 0.2 mas [17,18]. This data set contains nine GPS-type sources that already have redshift measurements. In this paper, we present the radio morphologies of the nine GPS sources at various pc-scale resolutions and report their core brightness temperatures, jet proper motions, viewing angles, variability, and radio spectra. Section 2 describes the sample selection and data collection. The observational results and discussion are presented in Section 3. The main results are then summarized in Section 4.

The cosmological parameters of $H_0 = 73$ km s$^{-1}$ Mpc$^{-1}$, $\Omega_M = 0.27$, and $\Omega_\Lambda = 0.73$ are adopted; 1 mas angular size corresponds to a projected linear size of 7.8 pc at $z = 1$.

## 2. Sample and Data

The target objects selected for this work were derived from the VLBI imaging survey of 134 compact AGNs at 43 GHz made by Cheng et al. [17,18]. The sources in their sample were selected by cross-matching the existing archival VLBI images with the Wilkinson Microwave Anisotropy Probe (MWAP) catalog and *Planck* catalog, for which no high-quality 43 GHz VLBI images have been published yet [19]. A typical *rms* noise in the 43 GHz VLBI image is 0.5 mJy beam$^{-1}$, allowing the reveal of compact jet structures at sub-milliarcsec (mas) scales.

We examined the radio spectra of these sources and finally found nine sources showing a convex radio spectrum, which is a typical feature of GPS objects. The basic information of these GPS sources, including two galaxies and seven quasars, is presented in Table 1.

The 43 GHz VLBA data were calibrated by Cheng et al. [18] in the NRAO package Astronomical Imaging Processing Software (AIPS 31DEC19 version, [20]). In addition to the 43 GHz VLBA data, the AIPS-calibrated VLBI data used in this paper were directly obtained from the Astrogeo database[1] at 8 GHz and from the Monitoring Of Jets in Active Galactic Nuclei with VLBA Experiments survey[2] (MOJAVE: [21,22]) at 15 GHz for a comprehensive analysis of the radio jet properties. We performed only a few iterations of self-calibration in Difmap software package [23] to eliminate the residual amplitude and phase errors. After self-calibration, the visibility data were fitted with several circular Gaussian components in Difmap using the MODELFIT program. The restoring beam is about $2.0 \times 1.0$ mas in the 8 GHz images, $1.5 \times 0.5$ in the 15 GHz images, and $0.5 \times 0.2$ in the 43 GHz images. The typical rms noise in the images is 0.5 mJy beam$^{-1}$, 0.2 mJy beam$^{-1}$, and 0.5 mJy beam$^{-1}$ at 8, 15, and 43 GHz, respectively. The representative total intensity images of the sources

are displayed in Figure 1. The total flux densities observed with single-dish telescopes or connected-element interferometers from the NASA/IPAC Extragalactic Database (NED)[3] are used to construct the radio spectra, which are shown in the right panel on each row of Figure 1. Error bars for the data points are not plotted for clarity of display.

**Table 1.** High frequency GPS sample.

| Name (1) | Other Name (2) | RA (J2000) (3) | Dec (J2000) (4) | z (5) | ID (6) | Ref. (7) | ID (8) |
|---|---|---|---|---|---|---|---|
| 0738 + 313 | OI + 363 | 07 41 10.70331 | + 31 12 00.2292 | 0.631 | QSO | M12 | |
| 0742 + 103 | PKS 0742 + 10 | 07 45 33.05952 | + 10 11 12.6922 | 2.624 | G | M12, S19 | HFP |
| 0743 − 006 | OI −072 | 07 45 54.08232 | −00 44 17.5399 | 0.994 | QSO | M12, S19 | HFP |
| 1124 − 186 | OM −148 | 11 27 04.39245 | −18 57 17.4418 | 1.048 | QSO | | |
| 2021 + 614 | OW + 637 | 20 22 06.68174 | + 61 36 58.8047 | 0.227 | G | M12, S19 | HFP |
| 2126 − 158 | OX −146 | 21 29 12.17590 | −15 38 41.0416 | 3.268 | QSO | M12, S19 | HFP |
| 2134 + 004 | DA 553 | 21 36 38.58630 | + 00 41 54.2129 | 1.941 | QSO | M12, S19 | HFP |
| 2209 + 236 | PKS 2209 + 236 | 22 12 05.96631 | + 23 55 40.5437 | 1.125 | QSO | M12 | |
| 2243 − 123 | PKS 2243 − 123 | 22 46 18.23198 | −12 06 51.2775 | 0.632 | QSO | | |

Note: these sources are selected from the sample of [18] with the characteristics of convex radio spectra. Column 6 gives the optical identification of the host galaxy: QSO—quasar; G—galaxy. References: M12—Mingaliev et al. [24]; S19—Sotnikova et al. [25]. Column 8 gives the radio spectrum identification: HFP—high-frequency peakers.

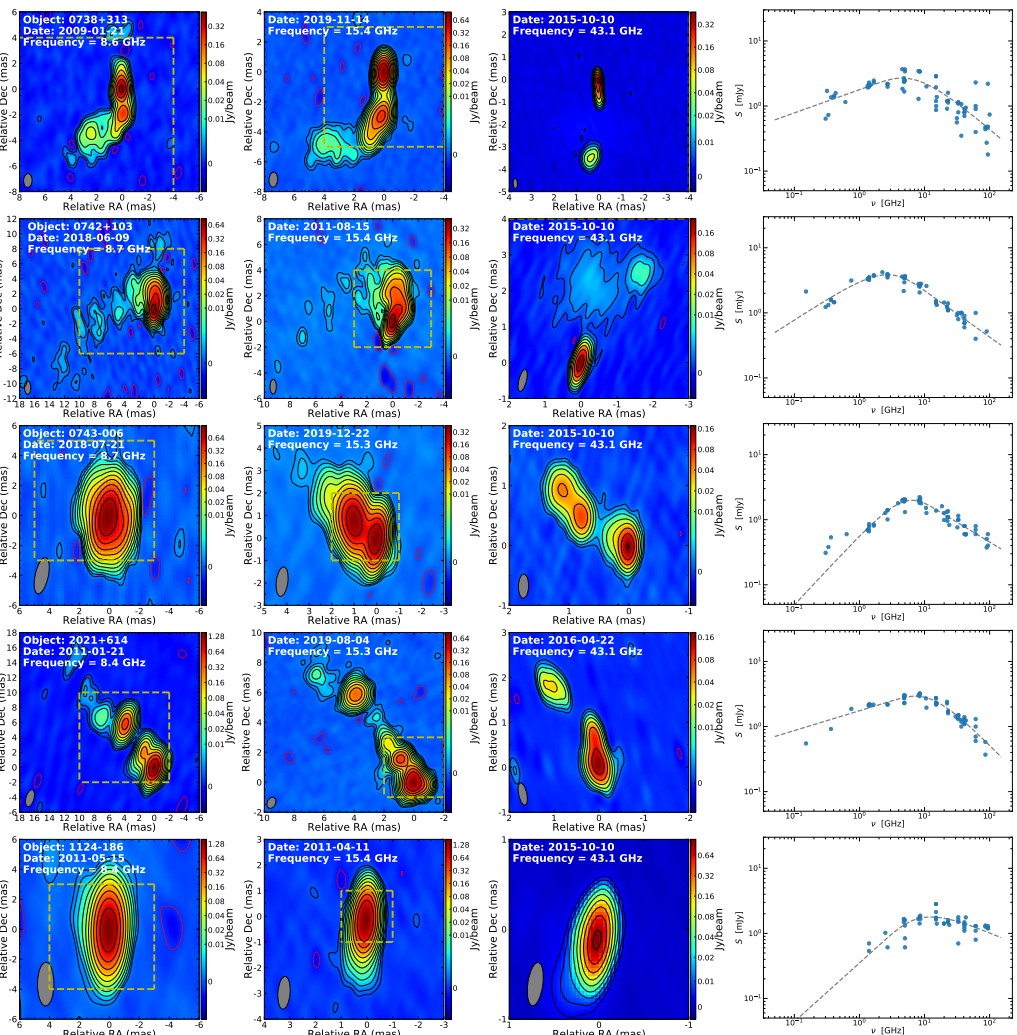

**Figure 1.** *Cont.*

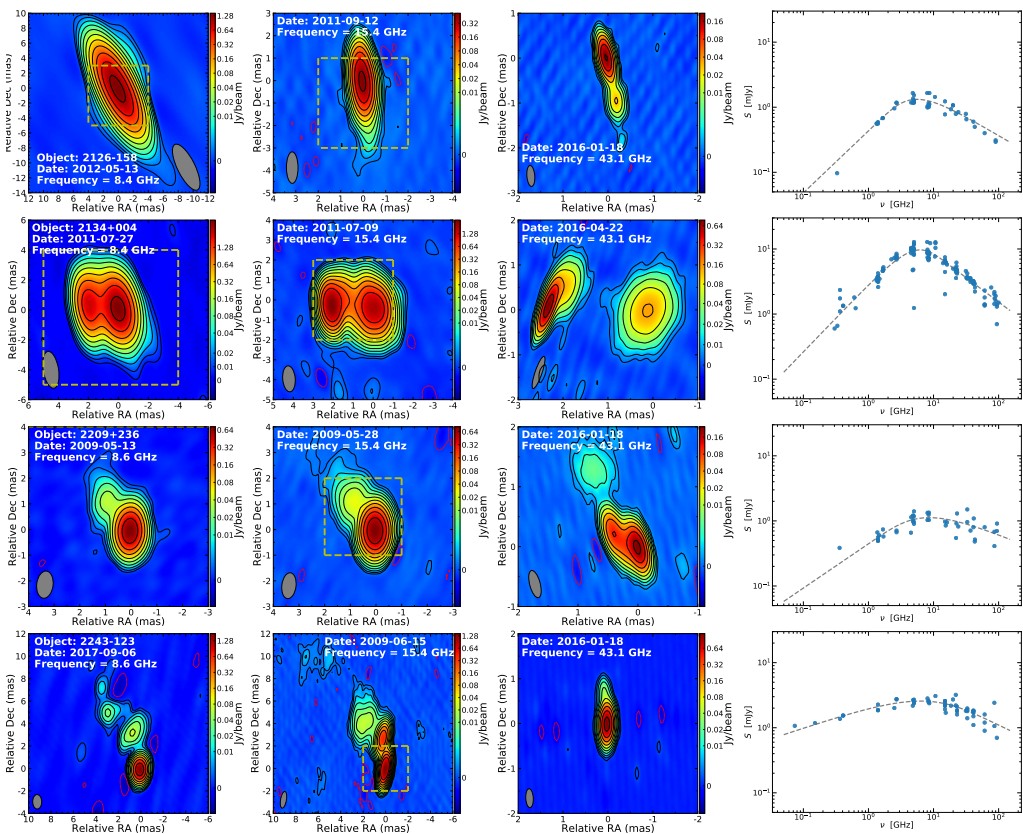

**Figure 1.** VLBI images at 8, 15, and 43 GHz and radio SEDs of the nine GPS sources. The peak intensity and lowest contour level ($\sim$3$\sigma$) are listed in Table 2. The contours increase in a step of 2. The grey-colored ellipse in the bottom-left corner of each panel denotes the restoring beam. The yellow-colored boxes indicate the image zone shown in higher-frequency images. The total flux densities of the entire source, measured by the connected interferometers or single dishes, are obtained from the NED and used to plot the radio spectrum. A function of the self-absorbed synchrotron radiation [15] is adopted to fit the radio spectrum data, shown as red-colored lines.

**Table 2.** VLBI image parameters.

| Name (1) | $\nu$ (GHz) (2) | $S_P$ (Jy beam$^{-1}$) (3) | $\sigma$ (mJy beam$^{-1}$) (4) | $\theta_{FWHM}$ (mas, mas, deg) (5) | $\alpha$ (6) |
|---|---|---|---|---|---|
| $0738 + 313$ | 8.6 | 0.545 | 0.61 | (1.09, 0.66, $-2.3$) | |
| | 15.4 | 0.988 | 0.14 | (1.00, 0.58, $-4.0$) | $-0.29$ |
| | 43.1 | 0.467 | 1.00 | (0.45, 0.19, 2.7) | $-0.56$ |
| $0742 + 103$ | 8.7 | 0.860 | 0.21 | (1.82, 0.84, 1.4) | |
| | 15.4 | 0.508 | 0.22 | (1.15, 0.50, $-1.3$) | $-0.51$ |
| | 43.1 | 0.257 | 0.50 | (0.62, 0.21, $-14.1$) | $-0.31$ |
| $0743 - 006$ | 8.7 | 1.041 | 0.25 | (1.93, 0.80, $-3.3$) | |
| | 15.3 | 0.455 | 0.07 | (1.32, 0.52, $-7.0$) | $-0.30$ |
| | 43.1 | 0.177 | 0.48 | (0.40, 0.17, $-1.4$) | $-0.18$ |
| $1124 - 186$ | 8.4 | 1.587 | 0.54 | (2.92, 1.10, $-0.4$) | |
| | 15.4 | 1.710 | 0.19 | (1.28, 0.47, $-4.4$) | $-0.31$ |
| | 43.1 | 1.100 | 1.20 | (0.49, 0.17, $-7.7$) | $-0.37$ |
| $2021 + 614$ | 8.4 | 1.510 | 0.49 | (2.33, 1.08, $-15.4$) | |
| | 15.3 | 0.709 | 0.07 | (0.78, 0.55, $-24.9$) | $-0.25$ |
| | 43.1 | 0.193 | 0.60 | (0.51, 0.19, 11.7) | $-1.25$ |
| $2126 - 158$ | 8.4 | 1.447 | 0.38 | (6.78, 2.34, 26.5) | |
| | 15.4 | 0.532 | 0.14 | (1.40, 0.52, 0.3) | $-0.53$ |
| | 43.1 | 0.219 | 0.35 | (0.47, 0.18, 4.6) | $-0.54$ |

**Table 2.** *Cont.*

| Name (1) | $\nu$ (GHz) (2) | $S_P$ (Jy beam$^{-1}$) (3) | $\sigma$ (mJy beam$^{-1}$) (4) | $\theta_{\mathrm{FWHM}}$ (mas, mas, deg) (5) | $\alpha$ (6) |
|---|---|---|---|---|---|
| $2134 + 004$ | 8.4 | 3.873 | 1.92 | (2.49, 1.02, 9.3) | |
| | 15.4 | 2.687 | 0.29 | (1.31, 0.63, 4.8) | 0.31 |
| | 43.1 | 0.779 | 1.50 | (0.75, 0.15, $-19.0$) | $-0.38$ |
| $2209 + 236$ | 8.6 | 0.721 | 0.54 | (1.07, 0.64, $-6.8$) | |
| | 15.4 | 0.822 | 0.14 | (1.01, 0.55, $-6.9$) | 0.07 |
| | 43.1 | 0.244 | 0.29 | (0.47, 0.18, 14.8) | $-0.68$ |
| $2243 - 123$ | 8.6 | 1.737 | 0.66 | (1.29, 0.80, 1.0) | |
| | 15.4 | 1.447 | 0.14 | (1.51, 0.52, $-9.8$) | $-0.04$ |
| | 43.1 | 1.190 | 0.86 | (0.42, 0.17, 3.0) | $-0.54$ |

Note column: 1—source name; 2—observing frequency, 3—peak intensity, 4—off-source rms noise in the CLEAN image; 5—restoring beam, 6—spectral index.

## 3. Results and Discussion

### 3.1. Radio Morphology

The first to third columns of Figure 1 show the naturally weighted total intensity images of the nine GPS sources at 8, 15, and 43 GHz. The 8 GHz images are obtained from the Astrogeo database; the 15 GHz images are from the MOJAVE project; and the 43 GHz images are reproduced from Cheng et al. [18]. The parameters for all individual images are listed in Table 2.

The identification of the core is crucial for morphology studies. With the high-resolution data at three frequencies (8, 15, and 43 GHz), we can easily obtain the spectral index of individual components to help identify the core, which is usually assumed to be a flat-spectrum component. In most cases of our sample, the core is located at one end of the jet, i.e., where the optical depth is unity ($\tau_\nu = 1$). Column 6 of Table 2 shows the spectral index of the core components identified in [26] at 8–15 GHz and 15–43 GHz. The data we used at 8 and 15 GHz were observed closest to the 43 GHz observation epoch. Four sources (J0745 + 1011, J0745 − 0044, J2022 + 6136, and J2136 + 0041) were found to show low levels of variability (<20%) with a constant spectral shape. However, the other five sources show a moderate level of variability (29–56%), so we may infer that the core identification may be an artifact of source variability. Based on the spectral index information, eight of the nine sources exhibit a compact core ($\alpha > -0.7$) and a one-sided jet structure, except for 2021 + 614, which may be a two-sided jet structure (i.e., a compact symmetric source) [27]. This suggests that 2021 + 614 has a large jet viewing angle (Table 3), which is consistent with its optical classification as a galaxy. Four sources (0743 − 006, 1124 − 186, 2126 − 158, and 2209 + 236) show a typical well-aligned core-jet structure. The recent new 43 GHz images show much finer structures within 2 mas, which are in good agreement with the results detected at low frequencies. Three sources (0738 + 313, 0742 + 103, and 2243 − 123) display a core-jet structure in the inner jet following a sharp bend in the out region (>2 mas). At 43 GHz, the core and the region near the bend are bright and fragmented into smaller features. The high-rotation measures of 0738 + 313 and 2243 − 123 near the bend suggest that the jet interacts with the interstellar medium [28]. The jet in 0742 + 103 does not exhibit brightening and high polarization near the bend, suggesting it may not be due to a collision [28].

The radio galaxy 2021 + 614 has been classified as a compact symmetric object [27]. There are several components within the 10 mas of the jet in the northeast-southwest direction. Two bright components are dominant at 8 and 15 GHz, with the radio core located between them, indicating a two-sided jet structure [27]. However, Lister [26] suggested that the core is located in the southernmost feature, which has a flat spectral index and is more variable and more compact, indicating a one-sided jet. The southernmost feature is well resolved in three components at 43 GHz [18]. The brightest component located at the end of the southwestern component (identified as the core in Lister [26]), but the middle component (labeled as J3 in Cheng et al. [18], identified as the core in Tschager et al. [27]), is more compact. We estimated the spectral indices of the component C and J3 by using the

15 GHz data in November 2017 [22] and our 43 GHz data in April 2016 despite possible variability. Components C and J3 show spectral indices of −1.25 and −0.33, respectively. We suggest that the more compact and flat spectrum of J3 should be associated with the central engine—the core, in good agreement with the previous identification of Tschager et al. [27].

**Table 3.** Radio jet parameters.

| Name | Mor. | $\beta_{\rm app}^{\rm max}$ | $T_{\rm b}$ $(10^{10}{\rm K})$ | $\delta$ | $\theta$ (deg) | $\nu_{\rm br}$ (GHz) | $\nu_{\rm p}$ (GHz) | $S_{\rm p}$ (Jy) | $B_{\rm eq}$ (mG) | V | $t_{\rm syn}$ (yr) |
|---|---|---|---|---|---|---|---|---|---|---|---|
| (1) | (2) | (3) | (4) | (5) | (6) | (7) | (8) | (9) | (10) | (11) | (12) |
| 0738 + 313 | c-j | 10.7 | 10.9 | 3.63 | 9.6 | >163 | 6.22 | 2.66 | 9.05 | 0.36 | <308 |
| 0742 + 103 | arch | ... | 8.4 | 2.80 | 29.3 | >326 | 10.42 | 3.84 | 15.57 | 0.21 | <92 |
| 0743 − 006 | c-j | 2.6 | 3.1 | 1.03 | 37.8 | >187 | 14.17 | 1.97 | 35.87 | 0.19 | <35 |
| 1124 − 186 | c-j | 8.3 | 3.8 | 1.27 | 13.6 | >193 | 25.10 | 1.78 | 25.62 | 0.56 | <58 |
| 2021 + 614 | C + Dj | 0.3 | 0.9 | 0.30 | 71.9 | >106 | 7.24 | 2.89 | 25.81 | 0.11 | <74 |
| 2126 − 158 | c-j | 4.0 | 0.7 | 0.23 | 29.0 | >384 | 23.13 | 1.32 | 48.24 | 0.42 | <15 |
| 2134 + 004 | arch | 4.8 | 14.9 | 4.97 | 11.6 | >294 | 19.60 | 9.64 | 21.02 | 0.11 | <65 |
| 2209 + 236 | c-j | 1.4 | 5.9 | 1.97 | 30.0 | >201 | 17.58 | 1.12 | 26.15 | 0.56 | <55 |
| 2243 − 123 | c-j | 4.9 | 2.2 | 0.73 | 22.4 | >153 | 11.13 | 2.57 | 15.37 | 0.29 | <139 |

Note column: 2—radio morphology, c-j—core-jet structure, C + Dj—core and double jets, Arch—an arch-like core jet structure, 3—the maximum measured jet speed, 4—the core brightness temperature at 43 GHz, 5—Doppler factors, 6—viewing angles, 7—break frequency, 8—peak frequency $\nu_{\rm p}$, 9—peak flux density $S_{\rm p}$, 10—equipartition magnetic field strength, 11—variability in flux density, 12—radiative age. References for column (3): apparent jet speed, [29–31]; column (4) core brightness temperature, Cheng et al. [18]; column (7) break frequency, Sotnikova et al. [25]; column (11) variability, Richards et al. [32].

The other GPS galaxy 0742 + 103 in this sample, despite demonstrating a one-sided core-jet structure, has undergone several oscillating changes in the direction of the jet within 10 mas, suggesting a helical jet pattern. It is one of the highest redshift GPS galaxies. Such a complex jet pattern is not rare in high-redshift radio-loud AGN [33]. The jet swings over a wide range of angles, lacking a fixed direction; it tends to easily lose mechanical energy and thus becomes unstable and confined in the host galaxy [34]. The jet of 0738 + 313 underwent a bend of more than 90 degrees at about 3–5 mas from the core, probably blocked by the interstellar medium.

The radio structure of 2134 + 004 is quite interesting. It looks similar to a CSO in the 8 and 15 GHz images, but the eastern component is brighter and more compact in the 43 GHz image than the western component. The radio spectrum of the whole source is a typical peaked spectrum with a turnover of around 20 GHz. The spectral index of the western component is −1.32 between 15 and 43 GHz; while the eastern component shows a rising spectrum with a peak above 15 GHz. The radio morphology and spectrum features of 2134 + 004 cannot distinguish between a core-jet or a CSO. Because it is optically identified as a $z \sim 2$ QSO, we are inclined to consider it as a core-jet source, in which the core is located in the eastern component ($\alpha = 0.31$ at 8–15 GHz and $\alpha = -0.38$ at 15–43 GHz).

### 3.2. Relativistic Beaming

The brightness temperature of the core component is a measure of the compactness of the radio emission and an indicator of relativistic beaming. We estimated the brightness temperature of the core component at 43 GHz and listed the values in Column 4 of Table 3. The median value of the core brightness temperature is $3.8 \times 10^{10}$ K, which is lower than the equipartition limit [35], showing no strong Doppler boosting. The core brightness temperatures derived at 43 GHz are systematically lower than those at 8 and 15 GHz [18,36,37]. The lower core brightness temperature and moderate Doppler factors suggest that the VLBI cores of GPS sources seen at 43 GHz may represent a jet region where the magnetic field energy dominates the total energy in the jet [35]. Doppler boosting is conventionally associated with relativistic jets seen at small viewing angles. We estimate the Doppler boosting factors as the ratio of $T_{\rm b}/T_{\rm int}$, where $T_{\rm b}$ is the core brightness temperature at 43 GHz and $T_{\rm int} = 3 \times 10^{10}$ K is the intrinsic brightness temperature [38]. The values of the

Doppler factors listed in column 5 of Table 3 range from 0.23 to 4.97. The median value of the Doppler factors at 43 GHz is 1.27, indicating mildly relativistic jets.

In the MOJAVE project, eight of the nine sources are monitored through 25 years, including multiple epoch observations [22]. Based on the measurements of these eight sources, a total of 27 jet components have been tracked for five or more individual epochs [29–31]. The maximum measured speeds for the eight sources, listed in column 3 of Table 3, show significant motions from slow apparent speed to superluminal motions, except for 2021 + 614. We find an obvious trend of increasing apparent component speed with increasing core distance in six sources, indicating that there may be an acceleration in the pc-scale radio jets.

The viewing angle $\theta$ can be calculated using the following relation:

$$\theta = \arctan \frac{2\beta}{\beta^2 + \delta^2 - 1} \tag{1}$$

For $\beta$ and $\delta$, we used the fastest measured radial non-accelerating apparent jet (column 3 of Table 3) and Doppler factor (column 5 of Table 3), respectively. For 0742 + 103, we can assume that the viewing angle of the jet is around the critical value $\theta_c = \arccos\beta$ for the maximal apparent speed at a given $\beta_{app}$. At this angle, the $\delta \sim \Gamma \sim \beta_{app} = 2.8$, and the intrinsic jet speed $\beta = \sqrt{1 - \frac{1}{\Gamma^2}}$ is approximately $0.87\,c$. We can obtain the viewing angle $\theta_c = 29.3°$. The viewing angle values listed in column 6 of Table 3 range from 9.6° to 71.9°. The median viewing angle is 22.4°, indicating that these GPS sources have large jet viewing angles, modest jet velocities, and low Doppler boosting coefficients. Comparing the Doppler factors and viewing angles in our sample with the well-monitored AGN by the Metsahovi Radio Observatory [39], we find that the observed Doppler factors are as low as those of galaxies, and the jet viewing angles are larger than those of most blazars.

### 3.3. Variability

In Figure 2, we show the radio light curves of the nine sources over a time span of 12 years from 2008 January to 2020 January 28 [32]. The GPS galaxies 0742 + 103 and 2021 + 614 (J2022 + 6136) show stable flux density throughout the whole monitoring period. The variability index, defined as $\frac{S_{max} - S_{min}}{S_{max} + S_{min}}$, is about 0.14 for 0742 + 103 and 0.11 for J2022 + 6136. 0742 + 103 (J0745 + 1011) seems to have entered an active period since mid-2019, and its flux density in the last observing epoch increased to $\sim$1.3 times the mean value. Continued monitoring of the flux density and radio structure changes can help to identify the correlation between variability and jet activity.

The GPS quasars in this sample show diverse variability properties. The variability index varies from 0.11 (J2136 + 0041) to 0.56 (J1127 − 1857, J2212 + 2355). The flux density of J2129 − 1538 shows a continuously increasing trend from 0.75 Jy to 1.75 Jy. We should note that the variability time scale of J2129 − 1538 in the observer's frame is elongated by the time dilation with a factor of $(1 + z)$ (i.e., ×4.268 for this source). J2136 + 0041 exhibits a stable flux density with a mean value of $6.83 \pm 0.29$ Jy but rapid variability on a time scale of $\sim$1 yr ($\sim$4 months in the source's rest frame). Prominent variabilities seen in J0741 + 3112, J1127 − 1857, J2212 + 2355, and J2246 − 1206 are consistent with the high beaming effect of their relativistic jets.

### 3.4. Spectrum and Age Estimates

Figure 1 displays the radio spectra of each source in the rightmost panels. The frequencies shown on *x*-axis have been corrected to the source's rest frame by multiplying a factor of $(1 + z)$. These spectra are made from the flux densities measured by connected-element interferometers and single-dish telescopes. Therefore, they represent the total flux densities of the entire sources. We caution that these are from non-simultaneous observations, as reflected in the scattering of the data points along the *y*-axis. A function of the self-absorbed synchrotron radiation is adopted to fit the radio spectrum data [15], shown as black lines. From the spectral fit, we obtained the rest-frame peak frequency $\nu_p$ and peak flux density

$S_p$, listed in columns 8 and 9 of Table 3. $\nu_p$ is in the range of 6.22–25.10 GHz with a mean peak frequency $\nu_p$ of 14.95 GHz, in good agreement with the typical GPS sources. $S_p$ ranges from 1.12 to 9.64 Jy with a mean flux density $S_p$ of 3.09 Jy.

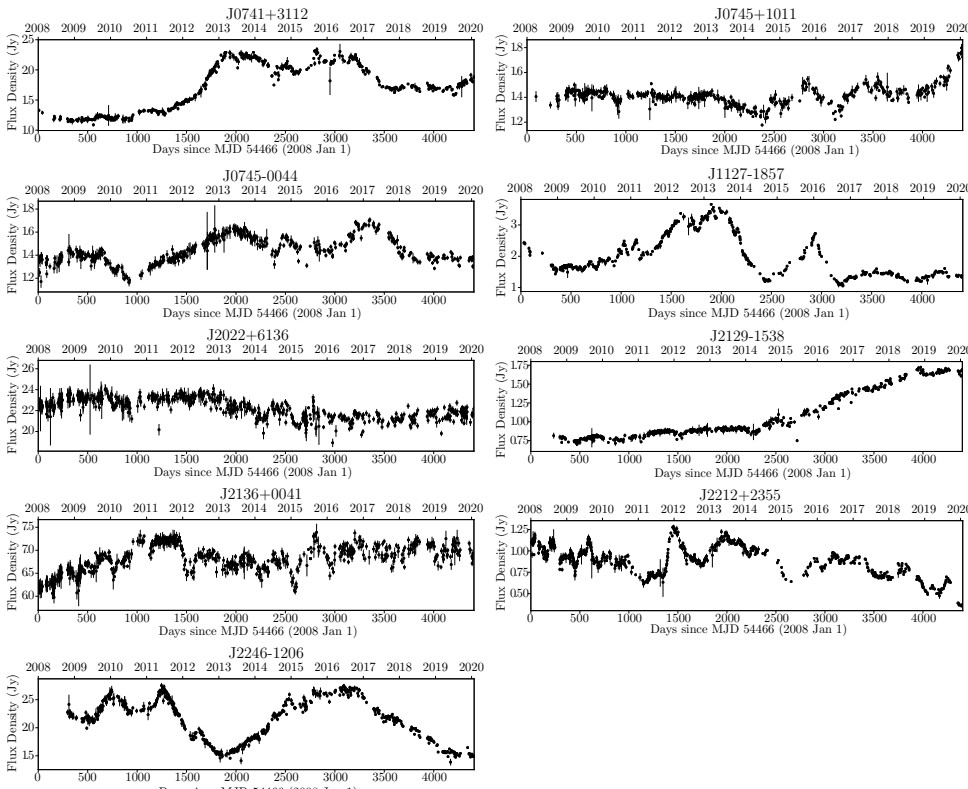

**Figure 2.** Fifteen-GHz radio light curves observed with the Owens Valley Radio Observatory (OVRO) [32].

The turnover frequency $\nu_p$ and peak flux density $S_p$ at the source's rest frame determined by the spectral fitting allow us to constrain the magnetic field in the emission region that dominates the flux density at the turnover frequency, assuming that the energy densities of particles and magnetic fields are in equipartition, following [40] as

$$B_{eq} = 123\eta^{2/7}(1+z)^{11/7}\left(\frac{d_L}{100\,\text{Mpc}}\right)^{-2/7}\left(\frac{\nu_p}{5\,\text{GHz}}\right)^{1/7}\left(\frac{S_p}{100\,\text{mJy}}\right)^{2/7}\left(\frac{\theta}{0.3''}\right)^{-6/7}\delta^{-5/7} \quad \mu\text{G}, \tag{2}$$

the parameters $d_L$, $\nu_p$, $S_p$, $\theta$, and $\delta$ are the luminosity distance in unit of 100 Mpc, the peak frequency in 5 GHz, the flux density at the peak frequency in Jy, the core size in mas (1.8 times the deconvolved Gaussian model size), and Doppler factor, respectively. $\eta$ is the ratio of energy density carried by protons and electrons to the energy density of the electrons. Here, we use $\eta = 1$ to obtain the minimum value of $B_{eq}$. The values of $B_{eq}$ are in the range of 9–48 mG as listed in column 10 of Table 3.

We cross-matched our sample with the RATAN-600 GPS catalog [25] resulting in an overlap of six GPS sources ($0738 + 313$, $0742 + 103$, $0743 − 006$, $2126−158$, $2134 + 004$, and $2209 + 236$). In the simultaneous observations of the RATAN-600 GPS catalog, we did not find a significant break in the high-frequency end of the spectrum, so the highest observed frequency of 22 GHz was chosen as the lower limit of the break frequency due to synchrotron aging. Combining with our 43 GHz data, we determined the break frequencies by fitting the CI mode using a broken-power law to the data [41]. The break frequency values in the source's rest frame are in the range of 106–384 GHz, listed in column 7 of Table 3. Due to the lack of sufficient high-frequency observing data (i.e., at frequencies higher than the turnover frequency), the current break frequencies are only used as a

lower limit. Future observations at higher frequencies would help to tightly constrain the break frequency.

Synchrotron radiation theory predicts that synchrotron losses preferentially deplete high-energy electron populations, leading to the radio spectrum steepening at high frequencies. It is possible to evaluate the age of the radiating electrons based on the break frequency and magnetic field strength. Under the assumption of the CI mode and the equipartition condition, we can express the radiative age of relativistic electrons as [42]:

$$t_{\rm syn} = 5.03 \times 10^4 \cdot B_{\rm eq}^{-1.5} \cdot \nu_{\rm br}^{-0.5} \qquad {\rm (yr)} \qquad (3)$$

where $B_{\rm eq}$ is the magnetic field strength in mG and $\nu_{\rm br}$ is the source's rest frame break frequency in GHz. We found that the radiative ages $t_{\rm syn}$ are in the range of $\sim$15–308 yr, which are much smaller than the typical young radio sources $10^2$–$10^5$ yr [2,42] and also the kinematic ages of CSOs [3,43]. We need to point out that the integrated spectrum is a mixture of spectra of different components, and the radiative age represents the lifetime of the electrons in the dominant emission region [42]. Because the sources are still compact and core-dominated at 43 GHz, the present results of the radiative age represent the lifetime of the electrons in the innermost core region and strongly support that these GPS sources represent a group of extremely young radio sources in a highly active state.

## 4. Summary

By combining our VLBA observations at 43 GHz with archival VLBI data at 8 and 15 GHz, one/two-sided jet structures have been detected for nine GPS sources on parsec scales. Four sources show a typical well-aligned core-jet structure; three sources display a core-jet in the inner jet following a sharp bend in the outer region (>2 mas). The jet bending in two sources is likely due to the interaction of the jet with the interstellar medium. In particular, 2021 + 614 has a symmetric double-components morphology at low frequency and is classified as a CSO based on the high-resolution images and spectra. The source 2134 + 004 displays an arched structure with a compact and bright core located on the southeastern end of the radio structure.

The core brightness temperatures of four sources are higher than the equipartition limit, inferring Doppler boosting factors ranging from 0.23 to 4.97 for relativistic jets. Jet viewing angles are estimated in the range of 9.6° to 71.9°. The GPS sources in this sample show diverse variability properties, ranging from 0.11 to 0.56.

We measured the peak frequency between 6.2 and 25.1 GHz in the source's rest frame and a flux density at the peak frequency, $S_{\rm m}$, ranging from 1.12 to 9.64 Jy. The equipartition magnetic field strengths range from 9 to 48 mG. Under the equipartition between particles and magnetic field energy densities, the spectral ages we found are in the range from $\sim$15–308 yr, supporting the hypothesis that these GPS sources are truly young radio sources.

More dedicated studies on a statistically complete sample of GPS sources with high-frequency VLBI observations are needed to enrich our understanding of the GPS source population and to contribute to presentation of a complete picture of radio galaxy evolution.

**Author Contributions:** Conceptualization, X.C. and T.A.; methodology, X.C.; software, X.C.; validation, A.W. and S.J.; formal analysis, X.C. and A.W.; investigation, X.C.; resources, X.C. and T.A.; data curation, X.C.; writing—original draft preparation, X.C.; writing—review and editing, T.A.; visualization, X.C.; supervision, T.A.; project administration, X.C.; funding acquisition, X.C. All authors have read and agreed to the published version of the manuscript.

**Funding:** This work was supported by the Brain Pool Program through the National Research Foundation of Korea (NRF) funded by the Ministry of Science and ICT (2019H1D3A1A01102564). T.A. thanks the Tianchi Talents program of Xinjiang.

**Data Availability Statement:** All observing data obtained by VLBA are archived via the Astrogeo website: http://astrogeo.org/.

**Acknowledgments:** The VLBA observations were sponsored by Shanghai Astronomical Observatory through an MoU with the NRAO (Project code: BA111). The Very Long Baseline Array is a facility of the National Science Foundation operated under cooperative agreement by Associated Universities, Inc. We acknowledge the use of data from the Astrogeo Center database maintained by Leonid Petrov. This research has made use of data from the MOJAVE database that is maintained by the MOJAVE team [22]. This research has made use of data from the OVRO 40-m monitoring program [32], supported by private funding from the California Insitute of Technology and the Max Planck Institute for Radio Astronomy, and by NASA grants NNX08AW31G, NNX11A043G, and NNX14AQ89G and NSF grants AST-0808050 and AST- 1109911. This work has made use of NASA Astrophysics Data System Abstract Service, and the NASA/IPAC Extragalactic Database (NED) which is operated by the Jet Propulsion Laboratory, California Institute of Technology, under contract with the National Aeronautics and Space Administration.

**Conflicts of Interest:** The authors declare no conflict of interest.

## Abbreviations

The following abbreviations are used in this manuscript:

| | |
|---|---|
| WMAP | Microwave anisotropy probe |
| MOJAVE | Monitoring Of Jets in Active galactic nuclei with VLBA Experiments |
| NED | NASA/IPAC Extragalactic Database |
| ISM | Interstellar medium |
| VLBI | Very-long-baseline interferometry |
| VLBA | Very-long-baseline array |
| CSO | Compact symmetric objects |
| GPS | GHz peaked spectrum |
| SED | Spectral energy distribution |
| QSO | Quasi-stellar object |
| mas | Milliarcsecond |

## Notes

1   The Astrogeo Database Maintained by L. Petrov. Available online: http://astrogeo.org/ (accessed on 15 December 2022).
2   MOJAVE. Available online: https://www.cv.nrao.edu/MOJAVE/index.html (accessed on 15 December 2022).
3   NED. Available online: http://ned.ipac.caltech.edu/ (accessed on 15 December 2022).

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
