# Peer review of "High-Frequency and High-Resolution VLBI Observations of GHz Peaked Spectrum Objects"

_galaxies, doi:10.3390/galaxies11020042_

Round 1

Reviewer 1 Report

Just some minor comments:

Page 1/ L21. Please add a couple of references to reinforce the final sentences of the paragraph.

Page 2/L44. Recently we reported--> Recently, we reported...

Page 3/Table 1. The abbreviation "HFD" was not previously introduced. It could be defined on the bottom notes of the table or somewhere else.

Page 3/Table 2. Please include a brief description of the Table and the parameters shown there (as in Tables 1 & 3).

Page 7/Fig 3. It is difficult to read the legends of this figure, too small.

Page 10/L267. In the list of abbreviations, please also include SED, QSO and 'mas'.

Reviewer 2 Report

The paper contains new observational data for a small sample of very interesting class of AGN - GPS (GHz-peaked spectrum objects). These active galactic nuclei revealing compact jets are thought to be very young. The paper content is interesting, impressive, and includes a complete set of radio-information for 9 GPS objects (radio maps at three frequencies, radio-spectra, and light curves at 15GHz measured during 12 years each). The presentation of the results is perfect. A faint irritation is only provoked by giving 'mean values' for all parameters estimated by the authors. The mean values are only justified for parameters with the normal probability distribution. Meantime, the sample of GPS objects analyzed is physically and evolutionarily inhomogeneous. For example, the GPS source 2021+614 is a low-redshift galaxy, with the symmetric two-side jet and the viewing jet angle of 72 deg; its characteristics are quite apart from others. Four one-sided jets are found to be straight and three - curved. Dealing with such inhomogeneous sample, it would be enough to give a range of parameters estimated for individual objects.

Author Response

We thank the referees for their comments and suggestions that improved the overall quality of this work.

Reviewer 3 Report

The paper presents analysis of subarcsec imaging of nine GPS sources, addressing ongoing debate if these type of sources are young one still within their host galaxy environment, or so-called frustrated ones, unable to leave the host galaxy dense medium. While the work presents some scientific merit, I think it needs to be justified a bit more to place it in the context of larger picture on the subject. My main criticism is the lack of further reaching conclusions of the presented analysis, and if there is none then the scientific value of the manuscript needs to be questioned. It looks like the sample is circumstantial, and it is unclear what it really say about the GPS population and the youth vs frustration debate; that is, how much the study of these nine sources improve our understanding or at least poses new questions. I think this needs to be addressed in a section "Conclusions" or "Discussion". Without this, the purpose of the paper is rather weak, and it is unclear what the point of the study is.

More specific comments:

1. Data: Throughout the paper it keeps getting unclear which data were processed by the authors and which were retrieved from literature/archives. Information on data processing would also be helpful. Also, please indicate the resolution of the data so that the discussion in section 3.1 can be properly verified.

2. All the figures should really be bigger as it is rather hard to read them at the moment. The markings on the images could also be thicker to be easily readable.

3. Figure 1: It is unclear which images are those of the authors and which ones are sourced from literature/archives.

4. Spectral indices: I find construction of the spectral indices in this study fundamentally flawed. Two major issues need to be addressed before these spectral indices can be used for the analysis.
a) Firstly, are the data compiled to derive the spectral indices compatible with each other? In other words, do they measure the same flux density or is there possibility that some of the measurements have so-called missing flux (partially resolved out)? Along this concern is also one of flux density scale and if the measurements were brought to a common one before further analysis.
b) Secondly, since the sources are variable, how robust is the spectral index calculation? Is it even possible to derive one within meaningful uncertainty?

5. In Section 3.4 it is confusing if the spectral are in the source's rest frame (2nd sentence of the section) or in the observed rest frame (2nd paragraph). By the way, in first paragraph "Caution that these are from...." is not a sentence (verb missing). Furthermore, in paragraph 3 of that section it is stated that there is lack of sufficient high-frequency observing data, while the whole point of the paper is to present the high-frequency observing data. 

Round 2

Reviewer 3 Report

I thank the authors for addressing my comments and concerns. I still feel that the figures are too small to be easily read, but I will leave that to the editors of the journal to discuss with the authors.